# Variance Reduction in Stochastic Gradient Langevin Dynamics

**Avinava Dubey**\*, **Sashank J. Reddi**\*, **Barnabás Póczos**, **Alexander J. Smola**, **Eric P. Xing**
Department of Machine Learning
Carnegie-Mellon University
Pittsburgh, PA 15213
{akdubey, sjakkamr, bapoczos, alex, epxing}@cs.cmu.edu

**Sinead A. Williamson**
IROM/Statistics and Data Science
University of Texas at Austin
Austin, TX 78712
sinead.williamson@mccombs.utexas.edu

## Abstract

Stochastic gradient-based Monte Carlo methods such as stochastic gradient Langevin dynamics are useful tools for posterior inference on large scale datasets in many machine learning applications. These methods scale to large datasets by using noisy gradients calculated using a mini-batch or subset of the dataset. However, the high variance inherent in these noisy gradients degrades performance and leads to slower mixing. In this paper, we present techniques for reducing variance in stochastic gradient Langevin dynamics, yielding novel stochastic Monte Carlo methods that improve performance by reducing the variance in the stochastic gradient. We show that our proposed method has better theoretical guarantees on convergence rate than stochastic Langevin dynamics. This is complemented by impressive empirical results obtained on a variety of real world datasets, and on four different machine learning tasks (regression, classification, independent component analysis and mixture modeling). These theoretical and empirical contributions combine to make a compelling case for using variance reduction in stochastic Monte Carlo methods.

## 1 Introduction

Monte Carlo methods are the gold standard in Bayesian posterior inference due to their asymptotic convergence properties; however convergence can be slow in large models due to poor mixing. Gradient-based Monte Carlo methods such as Langevin Dynamics and Hamiltonian Monte Carlo [10] allow us to use gradient information to more efficiently explore posterior distributions over continuous-valued parameters. By traversing contours of a potential energy function based on the posterior distribution, these methods allow us to make large moves in the sample space. Although gradient-based methods are efficient in exploring the posterior distribution, they are limited by the computational cost of computing the gradient and evaluating the likelihood on large datasets. As a result, stochastic variants are a popular choice when working with large data sets [15].

Stochastic gradient methods [13] have long been used in the optimization community to decrease the computational cost of gradient-based optimization algorithms such as gradient descent. These methods replace the (expensive, but accurate) gradient evaluation with a noisy (but computationally

cheap) gradient evaluation on a random subset of the data. With appropriate scaling, this gradient evaluated on a random subset of the data acts as a proxy for the true gradient. A carefully designed schedule of step sizes ensures convergence of the stochastic algorithm.

A similar idea has been employed to design stochastic versions of gradient-based Monte Carlo methods [15, 1, 2, 9]. By evaluating the derivative of the log likelihood on only a small subset of data points, we can drastically reduce computational costs. However, using stochastic gradients comes at a cost: While the resulting estimates are unbiased, they do have very high variance. This leads to an increased probability of selecting paths with high deviation from the true gradient, leading to slower convergence.

There have been a number of variations proposed on the basic stochastic gradient Langevin dynamics (SGLD) model of [15]: [4] incorporates a momentum term to improve posterior exploration; [6] proposes using additional variables to stabilize fluctuations; [12] proposes modifications to facilitate exploration of simplex; [7] provides sampling solutions for correlated data. However, none of these methods directly tries to reduce the variance in the computed stochastic gradient.

As was the case with the original SGLD algorithm, we look to the optimization community for inspiration, since high variance is also detrimental in stochastic gradient based optimization. A plethora of variance reduction techniques have recently been proposed to alleviate this issue for the stochastic gradient descent (SGD) algorithm [8, 5, 14]. By incorporating a carefully designed (usually unbiased) term into the update sequence of SGD, these methods reduce the variance that arises due to the stochastic gradients in SGD, thereby providing strong theoretical and empirical performance.

Inspired by these successes in the optimization community, we propose methods for reducing the variance in stochastic gradient Langevin dynamics. Our approach bridges the gap between the faster (in terms of iterations) convergence of non-stochastic Langevin dynamics, and the faster per-iteration speed of SGLD. While our approach draws its motivation from the stochastic optimization literature, it is to our knowledge the first approach that aims to directly reduce variance in a gradient-based Monte Carlo method. While our focus is on Langevin dynamics, our approach is easily applicable to other gradient-based Monte Carlo methods.

**Main Contributions:** We propose a new Langevin algorithm designed to reduce variance in the stochastic gradient, with minimal additional computational overhead. We also provide a memory efficient variant of our algorithm. We demonstrate theoretical conversion to the true posterior under reasonable assumptions, and show that the rate of convergence has a tighter bound than one previously shown for SGLD. We complement these theoretical results with empirical evaluation showing impressive speed-ups versus a standard SGLD algorithm, on a variety of machine learning tasks such as regression, classification, independent component analysis and mixture modeling.

## 2 Preliminaries

Let $\mathbf{X} = \{x_i\}_{i=1}^N$ be a set of data items modeled using a likelihood function $p(\mathbf{X}|\theta) = \prod_{i=1}^N p(x_i|\theta)$ where the parameter $\theta$ has prior distribution $p(\theta)$. We are interested in sampling from the posterior distribution $p(\theta|\mathbf{X}) \propto p(\theta) \prod_{i=1}^N p(x_i|\theta)$. If $N$ is large, standard Langevin Dynamics is not feasible due to the high cost of repeated gradient evaluations; a more scalable approach is to use a stochastic variant [15], which we will refer to as stochastic gradient Langevin dynamics, or SGLD. SGLD uses a classical Robbins-Monro stochastic approximation to the true gradient [13]. At each iteration $t$ of the algorithm, a subset $\mathbf{X}_t = \{x_{t1}, \ldots, x_{tn}\}$ of the data is sampled and the parameters are updated by using only this subset of data, according to

$$\Delta\theta_t = \frac{h_t}{2}\left(\nabla \log p(\theta_t) + \frac{N}{n}\sum_{i=1}^n \nabla \log p(x_{ti}|\theta_t)\right) + \eta_t \tag{1}$$

where $\eta_t \sim N(0, h_t)$, and $h_t$ is the learning rate. $h_t$ is set in such a fashion that $\sum_{t=1}^\infty h_t = \infty$ and $\sum_{t=1}^\infty h_t^2 < \infty$. This provides an approximation to a first order Langevin diffusion, with dynamics

$$d\theta = -\frac{1}{2}\nabla_\theta U dt + dW, \tag{2}$$

where $U$ is the unnormalized negative log posterior. Equation 2 has stationary distribution $\rho(\theta) \propto \exp\{-U(\theta)\}$. Let $\bar{\phi} = \int \phi(\theta)\rho(\theta)d\theta$ where $\phi$ represents a test function of interest. For a numerical

method that generates samples $\{\theta_t\}_{i=0}^{T-1}$, let $\hat{\phi}$ denote the empirical average $\frac{1}{T} \sum_{t=0}^{T-1} \phi(\theta_t)$. Furthermore, let $\psi$ denote the solution to the Poisson equation $\mathcal{L}\psi = \phi - \bar{\phi}$, where $\mathcal{L}$ is the generator of the diffusion, given by

$$\mathcal{L}\psi = \langle \nabla_\theta \psi, \nabla_\theta U \rangle + \frac{1}{2} \sum_i \nabla_i^2 \psi. \tag{3}$$

The decreasing step size $h_t$ in our approximation (Equation 1) means we do not have to incorporate a Metropolis-Hastings step to correct for the discretization error relative to Equation 2; however it comes at the cost of slowing the mixing rate of the algorithm. We note that, while the discretized Langevin diffusion is Markovian, its convergence guarantees rely on the quality of the approximation, rather than from standard Markov chain Monte Carlo analyses that rely on this Markovian property.

A second source of error comes from the use of stochastic approximations to the true gradients. This is equivalent to using an approximate generator $\tilde{\mathcal{L}}_t = \mathcal{L} + \Delta V_t$, where $\Delta V_t = (\nabla U_t - \nabla U) \cdot \nabla$ and $\nabla U_t$ is the current stochastic approximation to $\nabla U$. The key contribution of this paper will be replacing the Robbins-Monro approximation to $U$ with a lower-variance approximation, thus reducing the error.

To see more clearly the effect of the variance of our stochastic approximation on the estimator error, we present a result derived for SGLD by [3]:

**Theorem 1.** *[3] Let $U_t$ be an unbiased estimate of $U$ and $h_t = h$ for all $t \in \{1, \dots, T\}$. Then under certain reasonable assumptions (concretely, assumption [A1] in Section 4), for a smooth test function $\phi$, the MSE of SGLD at time $K = hT$ is bounded, for some $C > 0$ independent of $(T, h)$ in the following manner:*

$$\mathbb{E}(\hat{\phi} - \bar{\phi})^2 \le C \left( \underbrace{\frac{\frac{1}{T} \sum_t \mathbb{E}[\|\Delta V_t\|^2]}{T}}_{T_1} + \frac{1}{Th} + h^2 \right). \tag{4}$$

*Here $\|.\|$ represents the operator norm.*

We clearly see that the MSE depends on the variance term $\mathbb{E}[\|\Delta V_t\|^2]$, which in turn depends on the variance of the noisy stochastic gradients. Since, for consistency, we require $h \to 0$ as $T \to \infty$,[1] provided $\mathbb{E}[\|\Delta V_t\|^2]$ is bounded by a constant $\tau$, the term $T_1$ ceases to dominate as $T \to \infty$, meaning that the effect of noise in the stochastic gradient becomes negligible. However outside this asymptotic regime, the effect of the variance term in Equation 4 remains significant. This motivates our efforts in this paper to decrease the variance of the approximate gradient, while maintaining an unbiased estimator.

An easy to decrease the variance is by using larger minibatches. However, this comes at a considerable computational cost, undermining the whole benefit of using SGLD. Inspired by the recent success of variance reduction techniques in stochastic optimization [14, 8, 5], we take a rather different approach to reduce the effect of noisy gradients.

## 3 Variance Reduction for Langevin Dynamics

As we have seen in Section 2, reducing the variance of our stochastic approximation can reduce our estimation error. In this section, we introduce two approaches for variance reduction, based on recent variance reduction algorithms for gradient descent [5, 8]. The first algorithm, SAGA-LD, is appropriate when our bottleneck is computation; it yields improved convergence with minimal additional computational costs over SGLD. The second algorithm, SVRG-LD, is appropriate when our bottleneck is memory; while the computational cost is generally higher than SAGA-LD, the memory requirement is lower, with the memory overhead beyond that of stochastic Langevin dynamics scales as $O(d)$. In practice, we found that computation was a greater bottleneck in the examples considered, so our experimental section only focuses on SAGA-LD; however on larger datasets with easily computable gradients, SVRG-LD may be the optimal choice.

**Algorithm 1:** SAGA-LD

1: **Input:** $\alpha_0^i = \theta_0 \in \mathbb{R}^d$ for $i \in \{1, \ldots, N\}$, step sizes $\{h_t > 0\}_{i=0}^{T-1}$
2: $g_\alpha = \sum_{i=1}^N \nabla \log p(x_i|\alpha_0^i)$
3: **for** $t = 0$ **to** $T - 1$ **do**
4: $\quad$ Uniformly randomly pick a set $I_t$ from $\{1, \ldots, N\}$ (with replacement) such that $|I_t| = b$
5: $\quad$ Randomly draw $\eta_t \sim N(0, h_t)$
6: $\quad \theta_{t+1} = \theta_t + \frac{h_t}{2} \left( \nabla \log p(\theta_t) + \frac{N}{n} \sum_{i \in I_t} \left( \nabla \log p(x_i|\theta_t) - \nabla \log p(x_i|\alpha_t^i) \right) + g_\alpha \right) + \eta_t$
7: $\quad \alpha_{t+1}^i = \theta_t$ for $i \in I_t$ and $\alpha_{t+1}^i = \alpha_t^i$ for $i \notin I_t$
8: $\quad g_\alpha = g_\alpha + \sum_{i \in I_t} \left( \nabla \log p(x_i|\alpha_{t+1}^i) - \nabla \log p(x_i|\alpha_t^i) \right)$
9: **end for**
10: **Output:** Iterates $\{\theta_t\}_{t=0}^{T-1}$

## 3.1 SAGA-LD

The increased variance in SGLD is due to the fact that we only have information from $n \ll N$ data points at each iteration. However, inspired by a minibatch version of the SAGA algorithm [5], we can include information from the remaining data points via an approximate gradient, and partially update the average gradient in each operation. We call this approach SAGA-LD.

Under SAGA-LD, we explicitly store $N$ approximate gradients $\{g_{\alpha i}\}_{i=1}^N$, corresponding to the $N$ data points. Concretely, let $\alpha_t = (\alpha_t^i)_{i=1}^N$ be a set of vectors, initialized as $\alpha_0^i = \theta_0$ for all $i \in [N]$, and initialize $g_{\alpha i} = \nabla \log p(x_i|\alpha_0^i)$ and $g_\alpha = \sum_{i=1}^N g_{\alpha i}$. As we iterate through the data, if a data point is not selected in the current minibatch, we approximate its gradient with $g_{\alpha i}$. If $I_t = \{i_{1t}, \ldots i_{nt}\}$ is the minibatch selected at iteration $t$, this means we approximate the gradient as

$$\sum_{i=1}^N \nabla \log p(x_i|\theta_t) \approx \frac{N}{n} \sum_{i \in I_t} \left( \nabla \log p(x_i|\theta_t) - g_{\alpha i} \right) + g_\alpha \tag{5}$$

When Equation (5) is used for MAP estimation it corresponds to SAGA[5]. However by injecting noise into the parameter update in the following manner

$$\Delta \theta_t = \frac{h_t}{2} \left( \nabla \log p(\theta_t) + \frac{N}{n} \sum_{i \in I_t} \left( \nabla \log p(x_i|\theta_t) - g_{\alpha i} \right) + g_\alpha \right) + \eta_t, \text{ where } \eta_t \sim N(0, h_t) \tag{6}$$

we can adapt it for sampling from the posterior. After updating $\theta_{t+1} = \theta_t + \Delta \theta_t$, we let $\alpha_{t+1}^i = \theta_t$ for $i \in I_t$. Note that we do not need to explicitly store the $\alpha_t^i$; instead we just update the corresponding gradients $g_{\alpha i}$ and overall approximate gradient $g_\alpha$. The SAGA-LD algorithm is summarized in Algorithm 1.

The approximation in Equation (6) gives an unbiased estimate of the true gradient, since the minibatch $I_t$ is sampled uniformly at random from $[N]$, and the $\alpha_i^t$ are independent of $I_t$. SAGA-LD offers two key properties: (i) As shown in Section 4, SAGA-LD has better convergence properties than SGLD; (ii) The computational overhead is minimal, since SAGA-LD does not require explicit calculation of the full gradient. Instead, it simply makes use of gradients that are already being calculated in the current minibatch. Combined, we end up with a similar computational complexity to SGLD, with a much better convergence rate.

The only downside of SAGA-LD, when compared with SGLD, is in terms of memory storage. Since we need to store $N$ individual gradients $g_{\alpha i}$, we typically have a storage overhead of $O(Nd)$ relative to SGLD. Fortunately, in many applications of interest to machine learning, the cost can be reduced to $O(N)$ (please refer to [5] for more details), and in practice the cost of the higher memory requirements is typically outweighed by the improved convergence and low computational cost.

## 3.2 SVRG-LD

If the memory overhead of SAGA-LD is not acceptable, we can use a variant that reduces storage requirements, at the cost of higher computational demands. The memory complexity for SAGA-LD is high because the approximate gradient $g_\alpha$ is updated at each step. This can be avoided by updating the approximate gradient every $m$ iterations in a single evaluation, and never storing the individual gradients $g_{\alpha i}$. Concretely, after every $m$ passes through the data, we evaluate the gradient on the

**Algorithm 2:** SVRG-LD

---

1: **Input:** $\tilde{\theta} = \theta_0 \in \mathbb{R}^d$, epoch length $m$, step sizes $\{h_t > 0\}_{i=0}^{T-1}$
2: **for** $t = 0$ **to** $T - 1$ **do**
3:     **if** ($t \bmod m = 0$) **then**
4:         $\tilde{\theta} = \theta_t$
5:         $\tilde{g} = \sum_{i=1}^{N} \nabla \log p(x_i | \tilde{\theta})$
6:     **end if**
7:     Uniformly randomly pick a set $I_t$ from $\{1, \ldots, N\}$ (with replacement) such that $|I_t| = n$
8:     Randomly draw $\eta_t \sim N(0, h_t)$
9:     $\theta_{t+1} = \theta_t + \frac{h_t}{2} \left( \nabla \log p(\theta_t) + \frac{N}{n} \sum_{i \in I_t} \left( \nabla \log p(x_i | \theta_t) - \nabla \log p(x_i | \tilde{\theta}) \right) + \tilde{g} \right) + \eta_t$
10: **end for**
11: **Output:** Iterates $\{\theta_t\}_{t=0}^{T-1}$

---

entire data set, obtaining $\tilde{g} = \sum_{i=1}^{N} \tilde{g}_i$, where $\tilde{g}_i = \nabla \log p(x_i | \tilde{\theta})$ is the current local gradient. $\tilde{g}$ then serves as an approximate gradient until the next global evaluation. This yields an update of the form

$$\Delta\theta_t = \frac{h_t}{2} \left( \nabla \log p(\theta_t) + \frac{N}{n} \sum_{i \in I_t} (\nabla \log p(x_i | \theta_t) - \tilde{g}_i) + \tilde{g} \right) + \eta_t \text{ where } \eta_t \sim N(0, h_t) \quad (7)$$

Without adding noise $\eta_t$ the update sequence in Equation (7) corresponds to the stochastic variance reduction gradient descent algorithm [8]. Pseudocode for this procedure is given in Algorithm 2.

While the memory requirements are lower, the computational cost is higher, due to the cost of a full update of $\tilde{g}$. Further, convergence may be negatively effected due to the fact that, as we move further from $\tilde{\theta}$, $\tilde{g}$ will be further from the true gradient. In practice, we found SAGA-LD to be a more effective algorithm on the datasets considered, so in the interest of space we relegate further details about SVRG-LD to the appendix.

## 4 Analysis

Our motivation in this paper was to improve the convergence of SGLD, by reducing the variance of the gradient estimate. As we saw in Theorem 1, a high variance $\mathbb{E}[||\Delta V_t||^2]$, corresponding to noisy stochastic gradients, leads to a large bound on the MSE of a test function. We expand this analysis to show that the algorithms introduced in this paper yield a tighter bound.

Theorem 1 required a number of assumptions, given below in [A1]. Discussion of the reasonableness of these assumptions is provided in [3].

**[A1]** We assume the functional $\psi$ that solves the Poisson equation $\mathcal{L}\psi = \phi - \bar{\phi}$ is bounded up to 3rd-order derivatives by some function $\Gamma$, i.e., $\|\mathcal{D}^k \psi\| \leq C_k \Gamma^{p_k}$ where $\mathcal{D}$ is the $k$th order derivative (for $k = (0, 1, 2, 3)$), and $C_k, p_k > 0$. We also assume that the expectation of $\Gamma$ on $\{\theta_t\}$ is bounded ($\sup_t \mathbb{E}\Gamma^p[\theta_t] < \infty$) and that $\Gamma$ is smooth such that $\sup_{s \in (0,1)} \Gamma^p(s\theta + (1-s)\theta') \leq C(\Gamma^p(\theta) + \Gamma^p(\theta'))$, $\forall \theta, \theta', p \leq \max 2p_k$ for some $C > 0$.

In our analysis of SAGA-LD and SVRG-LD, we make the assumptions in [A1], and add the following further assumptions about the smoothness of our gradients:

**[A2]** We assume that the functions $\log p(x_i | \theta)$ are Lipschitz smooth with constant $L$ for all $i \in [N]$, i.e. $\|\nabla \log p(x_i | \theta) - \nabla \log p(x_i | \theta')\| \leq L \|\theta - \theta'\|$ for all $i \in [N]$ and $\theta, \theta' \in \mathbb{R}^d$. We assume that $(\Delta V_t \psi(\theta))^2 \leq C' \|\nabla U_t(\theta) - \nabla U(\theta)\|^2$ for some constant $C' > 0$ for all $\theta \in \mathbb{R}^d$, where $\psi$ is the solution to the Poisson equation for our test function. We also assume that $\|\nabla \log p(\theta)\| \leq \sigma$ and $\|\nabla \log p(x_i | \theta)\| \leq \sigma$ for some $\sigma$ and all $i \in [N]$ and $\theta \in \mathbb{R}^d$.

The Lipschitz smoothness assumption is very common both in the optimization literature [11] and when working with Itô diffusions [3]. The bound on $(\Delta V_t \psi(\theta))^2$ holds when the gradient $\|\nabla\psi\|$ is bounded.

Loosely, these assumptions encode the idea that the gradients don't change too quickly, so that we limit the errors introduced by incorporating gradients based on previous values of $\theta$. With these assumptions, we state the following key results for SAGA-LD and SVRG-LD, which are proved in the supplement.

**Theorem 2.** *Let $h_t = h$ for all $t \in \{1, \ldots, T\}$. Under the assumptions [A1],[A2], for a smooth test function $\phi$, the MSE of* SAGA-LD *(in Algorithm 1) at time $K = hT$ is bounded, for some $C > 0$ independent of $(T, h)$ in the following manner:*

$$\mathbb{E}(\hat{\phi} - \bar{\phi})^2 \leq C \left( \frac{N^2 \min\{\sigma^2, \frac{N^2}{n^2}(L^2 h^2 \sigma^2 + hd)\}}{nT} + \frac{1}{Th} + h^2 \right). \tag{8}$$

A similar result can be shown for SVRG-LD in Algorithm 2:

**Theorem 3.** *Let $h_t = h$ for all $t \in \{1, \ldots, T\}$. Under the assumptions [A1],[A2], for a smooth test function $\phi$, the MSE of* SVRG-LD *(in Algorithm 2) at time $K = hT$ is bounded, for some $C > 0$ independent of $(T, h)$ in the following manner:*

$$\mathbb{E}(\hat{\phi} - \bar{\phi})^2 \leq C \left( \frac{N^2 \min\{\sigma^2, m^2(L^2 h^2 \sigma^2 + hd)\}}{nT} + \frac{1}{Th} + h^2 \right). \tag{9}$$

The result in Theorem 3 is qualitatively equivalent to that in Theorem 2 when $m = \lfloor N/n \rfloor$. In general, such a choice of $m$ is preferable because, in this case, the overall cost of calculation of full gradient in Algorithm 2 becomes insignificant.

In order to assess the theoretical convergence of our proposed algorithm, we compare the bounds for SVRG-LD (Theorem 3) and SAGA-LD (Theorem 2) with those obtained for SGLD (Theorem 1. Under the assumptions in this section, it is easy to show that the term $T_1$ in Theorem 1 becomes $O(N^2\sigma^2/(Tn))$. In contrast, both Theorem 2 and 3 show that, due to a reduction in variance, SVRG-LD and SAGA-LD exhibit a much weaker dependence. More specifically, this is manifested in the form of the following bound:

$$\frac{N^2 \min\left\{\sigma^2, \frac{N^2}{n^2}(h^2\sigma^2 + hd)\right\}}{nT}.$$

Note that this is tighter than the corresponding bound on SGLD. We also note that, similar to SGLD, SAGA-LD and SVRG-LD require $h \to 0$ as $T \to \infty$. In such a scenario, the convergence becomes significantly faster relative to SGLD as $h \to 0$.

## 5 Experiments

We present our empirical results in this section. We focus on applying our stochastic gradient method to four different machine learning tasks, carried out on benchmark datasets: (i) Bayesian linear regression (ii) Bayesian logistic regression and (iii) Independent component analysis (iv) Mixture modeling. We focus on SAGA-LD, since in the applications considered, the convergence and computational benefits of SAGA-LD are more beneficial than the memory benefits of SVRG-LD;

In order to reduce the initial computational costs associated with calculating the initial average gradient, we use a variant of Algorithm 1 that calculates $g_\alpha$ (in line 2 of Algorithm 1) in an online fashion and reweights the updates accordingly. Note that such a heuristic is also commonly used in the implementation of SAG and SAGA in the context of optimization [14, 5].

In all our experiments, we use a decreasing step size for SGLD as suggested by [15]. In particular, we use $\epsilon_t = a(b + t)^{-\gamma}$, where the parameters $a$, $b$ and $\gamma$ are chosen for each dataset to give the best performance of the algorithm on that particular dataset. For SAGA-LD, due to the benefit of variance reduction, we use a simple two phase constant step size selection strategy. In each of these phases, a constant step size is chosen such that SAGA-LD gives the best performance on the particular dataset. The minibatch size, $n$, in both SGLD and SAGA-LD is held at a constant value of 10 throughout our experiments. All algorithms are initialized to the same point and the same sequence of minibatches is pre-generated and used in both algorithms.

### 5.1 Regression

We first demonstrate the performance of our algorithm on Bayesian regression. Formally, we are provided with inputs $\mathbf{Z} = \{x_i, y_i\}_{i=1}^N$ where $x_i \in \mathbb{R}^d$ and $y_i \in \mathbb{R}$. The distribution of the $i^{\text{th}}$ output $y_i$ is given by $p(y_i|x_i) = \mathcal{N}(\beta^\top x_i, \sigma_e)$, where $p(\beta) = \mathcal{N}(0, \lambda^{-1}I)$. Due to conjugacy, the posterior distribution over $\beta$ is also normal, and the gradients of the log-likelihood and the log-prior are given

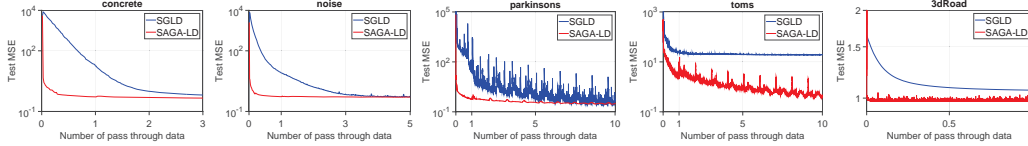

Figure 1: Performance comparison of SGLD and SAGA-LD on a regression task. The x-axis and y-axis represent the number of passes through the entire data and the average test MSE, respectively. Additional experiments are provided in the appendix.

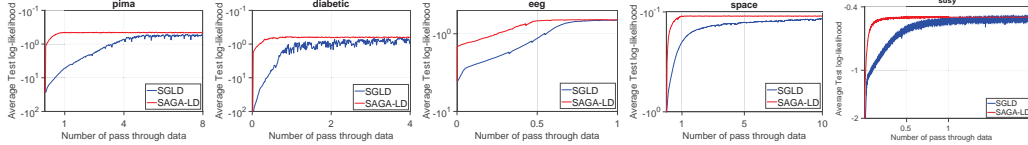

Figure 2: Comparison of performance of SGLD and SAGA-LD for Bayesian logistic regression. The x-axes and y-axes represent the number of effective passes through the dataset and the test log-likelihood, respectively.

by $\nabla_\beta \log(P(y_i|x_i, \beta)) = -(y_i - \beta^T x_i)x_i$ and $\nabla_\beta \log(P(\beta)) = -\lambda\beta$. We ran experiments on 11 standard UCI regression datasets, summarized in Table 1.[2] In each case, we set the prior precision $\lambda = 1$, and we partitioned our dataset into training (70%), validation (10%), and test (20%) sets. The validation set is used to select the step size parameters, and we report the mean square error (MSE) evaluated on the test set, using 5-fold cross-validation.

The average test MSE on a subset of datasets is reported in Figure 1. Due to space constraints, we relegate the remaining experimental results to the appendix. As shown in Figure 1, SAGA-LD converges much faster than the SGLD method (taking less than one pass through the whole dataset in many cases). This performance gain is consistent across all the datasets. Furthermore, the step size selection was much simpler for SAGA-LD than SGLD.

| Datasets | concrete | noise | parkinson | bike | toms | protein | casp | kegg | 3droad | music | twitter |
|---|---|---|---|---|---|---|---|---|---|---|---|
| N | 1030 | 1503 | 5875 | 17379 | 45730 | 45730 | 53500 | 64608 | 434874 | 515345 | 583250 |
| P | 8 | 5 | 21 | 12 | 96 | 9 | 9 | 27 | 2 | 90 | 77 |

Table 1: Summary of datasets used for regression.

## 5.2 Classification

We next turn our attention to classification, using Bayesian logistic regression. In this case, the input is the set $\mathbf{Z} = \{x_i, y_i\}_{i=1}^N$ where $x_i \in \mathbb{R}^d$, $y_i \in \{0, 1\}$. The distribution of the output $y_i$ for given sample $x_i$ is given by $P(y_i = 1) = \phi(\beta^T x_i)$, where $p(\beta) = \mathcal{N}(0, \lambda^{-1}I)$ and $\phi(z) = 1/(1 + exp(-z))$. Here, the gradient of the log-likelihood and the log-prior are given by $\nabla_\beta \log(P(y_i|x_i, \beta)) = (y_i - \phi(\beta^T x_i))x_i$ and $\nabla_\beta \log(P(\beta)) = -\lambda\beta$ respectively. Again, $\lambda$ is set to 1 for all experiments, and the dataset split and parameter selection method is exactly same as in our regression experiments. We run experiments on five binary classification datasets in the UCI repository, summarized in Table 2, and report the the test set log-likelihood for each dataset, using 5-fold cross validation. Figure 2 shows the performance of SGLD and SAGA-LD for the classification datasets. As we saw with the regression task, SAGA-LD converges faster that SGLD on all the datasets, demonstrating the efficiency of the our algorithm in this setting.

| Datasets | pima | diabetic | eeg | space | susy |
|---|---|---|---|---|---|
| N | 768 | 1151 | 14980 | 58000 | 100000 |
| d | 8 | 20 | 15 | 9 | 18 |

Table 2: Summary of the datasets used for classification.

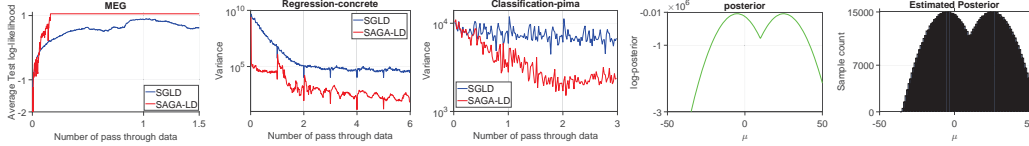

Figure 3: The left plot shows the performance of SGLD and SAGA-LD for the ICA task. The next two plots show the variance of SGLD and SAGA-LD for regression and classification. The rightmost two plot shows true and estimated posteriors using SAGA-LD for the mixture modeling task

## 5.3 Bayesian Independent Component Analysis

To evaluate performance under a Bayesian Independent Component Analysis (ICA) model, we assume our dataset $\mathbf{x} = \{x_i\}_{i=1}^{N}$ is distributed according to

$$p(\mathbf{x}|W) \propto |\det(W)| \prod_{i=1}^{d} p(y_i), \quad W_{ij} \sim \mathcal{N}(0, \lambda), \tag{10}$$

where $W \in \mathbb{R}^{d \times d}$, $y_i = w_i^T x$, and $p(y_i) = 1/(4\cosh^2(\frac{1}{2}y_i))$. The gradient of the log-likelihood and the log-prior are $\nabla_W \log(p(x_i|W)) = (W^{-1})^T - Y_i x_i^T$ where $Y_{ij} = \tanh(\frac{1}{2}y_{ij})$ for all $j \in [d]$ and $\nabla_W \log(p(W)) = -\lambda W$ respectively. All other parameters are set as before. We used a standard ICA dataset for our experiment[3], comprisein 17730 time-points with 122 channels from which we extracted the first 10 channels. Further experimental details are similar to those for regression and classification. The performance (in terms of test set log likelihood) of SGLD and SAGA-LD for the ICA task is shown in Figure 3. As seen in Figure 3, similar to the regression and classification tasks, SAGA-LD outperforms SGLD in the ICA task.

## 5.4 Mixture Model

Finally, we evaluate how well SAGA-LD estimates the true posterior of parameters of mixture models. We generated 20,000 data points from a mixture of two Gaussians, given by $p(x|\mu, \sigma_1, \sigma_2, \gamma) = \frac{1}{2}\mathcal{N}(x; \mu, \sigma^2) + \frac{1}{2}\mathcal{N}(x; -\mu + \gamma, \sigma^2)$, where $\mu = -5$, $\gamma = 20$, and $\sigma = 5$. We estimate the posterior distribution over $\mu$, holding the other variables fixed. The two plots on the right of Figure 3 show that we are able to estimate the true posterior correctly.

**Discussion**: Our experiments provide a very compelling reason to use variance reduction techniques for SGLD, complementing the theoretical justification given in Section 4. The hypothesized variance reduction is demonstrated in Figure 3, where we compare the variances of SGLD and SAGA-LD with respect to the true gradient on regression and classification tasks. As we see from all of the experimental results in this section, SAGA-LD converges with relatively very few samples compared with SGLD. This is especially important in hierarchical Bayesian models where, typically, the size of the model used is proportional to the number of observations. Thus, with SAGA-LD, we can achieve better performance with very few samples. Another advantage is that, while we require the step size to tend to zero, we can use a much simpler schedule than SGLD.

## 6 Discussion and Future Work

SAGA-LD is a new stochastic Langevin method that obtains improved convergence by reducing the variance in the stochastic gradient. An alternative method, SVRG-LD, can be used when memory is at a premium. For both SAGA-LD and SVRG-LD, we proved a tighter convergence bound than the one previously shown for stochastic gradient Langevin dynamics. We also showed, on a variety of machine learning tasks, that SAGA-LD converges to the true posterior faster than SGLD, suggesting the widespread use of SAGA-LD in place of SGLD.

We note that, unlike other stochastic Langevin methods, our sampler is non-Markovian. Since our convergence guarantees are based on bounding the error relative to the full Langevin diffusion rather than on properties of a Markov chain, this does not impact the validity of our sampler.

While we showed the efficacy of using our proposed variance reduction technique to SGLD, our proposed strategy is very generic enough and can also be applied to other gradient-based MCMC techniques such as [1, 2, 9, 6, 12]. We leave this as future work.

## Footnotes

\* denotes equal contribution

[1] In particular, if $h \propto T^{-1/3}$, we obtain the optimal convergence rate for the above upper bound.

[2]The datasets can be downloaded from `https://archive.ics.uci.edu/ml/index.html`

[3]The dataset can be downloaded from `https://www.cis.hut.fi/projects/ica/eegmeg/MEG_data.html`.

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
