[Supplementary Material]

# Appendix - Variance Reduction in Stochastic Gradient Langevin Dynamics

## Appendix

In this appendix, we provide details of the theoretical results given in the main paper. We first start with the proof of Theorem 3 and then look at the proof of Theorem 2.

We also include some additional experimental results for the regression experiments.

## A   Proof of Theorem 3

We introduce a notation for simplifying our theoretical exposition. For any $t \in \{sm, (s+1)m\}$ for some integer $s \in \{0, \lfloor T/m \rfloor\}$, let $\tilde{\theta}_t = \theta_{sm}$. Then the SVRG-LD update can be rewritten as

$$\theta_{t+1} = \theta_t + \frac{h_t}{2}\left(\nabla \log p(\theta_t) + \frac{N}{n}\sum_{i \in I_t}(\nabla \log p(x_i|\theta_t) - \nabla \log p(x_i|\tilde{\theta}_t)) + \sum_{i=1}^{N}\nabla \log p(x_i|\tilde{\theta}_t)\right) + \eta_t.$$

To prove Theorem 3, we start from a bound proved in [3] for general stochastic gradient MCMC. First, recall that $\hat{\phi} = \frac{1}{T}\sum_t \phi(\theta_t)$ is the empirical average of some smooth test function $\phi$, and that MSE of the average $\bar{\phi}$ of $\phi$ is $\mathbb{E}(\hat{\phi} - \bar{\phi})^2$. Then [3] show that, for SGLD,

$$\mathbb{E}(\phi - \bar{\phi})^2 \leq C\left(\frac{\frac{1}{T}\sum_t \mathbb{E}(\Delta V_t \psi(\theta_t))^2}{T} + \frac{1}{Th} + h^2\right). \tag{11}$$

for some constant $C > 0$.

[3] then use an upper bound on the term $\frac{1}{T}\sum_t \mathbb{E}(\Delta V_t \psi(\theta_t))^2$ to tighten this bound, under the conditions in Assumption [A1], to obtain the result in Theorem 1.

We use a different upper bound for the term $\frac{1}{T}\sum_t \mathbb{E}(\Delta V_t \psi(\theta_t))^2$. For some constant $C'$, we have:

$$\frac{1}{C'T}\sum_t \mathbb{E}(\Delta V_t \psi(\theta_t))^2 \leq \frac{1}{T}\sum_t \mathbb{E}[\|\nabla U_t(\theta_t) - \nabla U(\theta_t)\|^2]$$

$$= \frac{1}{T}\sum_t \mathbb{E}\left\|\frac{N}{n}\sum_{i \in I_t}(\nabla \log p(x_i|\theta_t) - \nabla \log p(x_i|\tilde{\theta}_t)) + \sum_{i=1}^{N}\nabla \log p(x_i|\tilde{\theta}_t) - \sum_{i=1}^{N}\nabla \log p(x_i|\theta_t)\right\|^2$$

$$= \frac{1}{Tn^2}\sum_t \mathbb{E}\left\|\sum_{i \in I_t}\left(N\left[\nabla \log p(x_i|\theta_t) - \nabla \log p(x_i|\tilde{\theta}_t)\right] - \left[\sum_{i=1}^{N}\nabla \log p(x_i|\theta_t) - \sum_{i=1}^{N}\nabla \log p(x_i|\tilde{\theta}_t)\right]\right)\right\|^2$$

$$= \frac{1}{Tn^2}\sum_t \mathbb{E}\sum_{i \in I_t}\left\|\left(N\left[\nabla \log p(x_i|\theta_t) - \nabla \log p(x_i|\tilde{\theta}_t)\right] - \left[\sum_{i=1}^{N}\nabla \log p(x_i|\theta_t) - \sum_{i=1}^{N}\nabla \log p(x_i|\tilde{\theta}_t)\right]\right)\right\|^2$$

$$\leq \frac{1}{Tn^2}\sum_t \mathbb{E}\sum_{i \in I_t}\left\|N\left[\nabla \log p(x_i|\theta_t) - \nabla \log p(x_i|\tilde{\theta}_t)\right]\right\|^2 \leq \frac{L^2 N^2}{Tn}\sum_t \mathbb{E}\left\|\theta_t - \tilde{\theta}_t\right\|^2. \tag{12}$$

The first inequality follows from our assumption [A3] in Section 4. The third equality follows from Lemma 1. The second inequality is due to the fact that $\mathbb{E}[\|\zeta - \mathbb{E}[\zeta]\|^2] \leq \mathbb{E}[\|\zeta\|^2]$ for any random variable $\zeta \in \mathbb{R}^d$. The last inequality is follows from the Lipschitz continuity of $\nabla \log p(x_i|\theta)$. Note that alternatively, we can also bound in the following fashion:

$$\frac{1}{C'T}\sum_t \mathbb{E}(\Delta V_t \psi(\theta_t))^2 \leq \frac{1}{Tn^2}\sum_t \mathbb{E}\sum_{i \in I_t}\left\|N\left[\nabla \log p(x_i|\theta_t) - \nabla \log p(x_i|\tilde{\theta}_t)\right]\right\|^2 \leq \frac{2N^2 \sigma^2}{n}$$

$$\tag{13}$$

Consider $t \in \{sm + 1, (s+1)m\}$ for some integer $s \in \{0, \lfloor T/m \rfloor\}$. We bound $\mathbb{E}\|\theta_t - \tilde{\theta}\|^2$ in the following manner:

$$\mathbb{E}\left\|\theta_t - \tilde{\theta}_t\right\|^2 = \mathbb{E}\left\|\sum_{j=sm}^{t-1}(\theta_{j+1} - \theta_j)\right\|^2$$

$$\leq (t - sm)\sum_{j=sm}^{t-1}\mathbb{E}[\|\theta_{j+1} - \theta_j\|^2] \leq m\sum_{j=sm}^{t-1}\mathbb{E}[\|\theta_{j+1} - \theta_j\|^2]. \tag{14}$$

The first inequality is due to Lemma 2. The second inequality is due to the fact that $t \in \{sm + 1, (s+1)m\}$. We bound the term $\mathbb{E}[\|\theta_{j+1} - \theta_j\|^2]$ in the following manner:

$$\mathbb{E}[\|\theta_{j+1} - \theta_j\|^2]$$

$$= E\left\|\frac{h}{2}\left(\nabla \log p(\theta_j) + \frac{N}{n}\sum_{i \in I_t}(\nabla \log p(x_i|\theta_j) - \nabla \log p(x_i|\tilde{\theta}_j)) + \sum_{i=1}^{N}\nabla \log p(x_i|\tilde{\theta}_j)\right) + \eta_j\right\|^2$$

$$\leq \frac{3h^2}{4}E\|\nabla \log p(\theta_j)\|^2 + 3\mathbb{E}[\|\eta_j\|^2]$$

$$+ \frac{3h^2}{4}\left\|\frac{N}{n}\sum_{i \in I_t}(\nabla \log p(x_i|\theta_j) - \nabla \log p(x_i|\tilde{\theta}_j)) + \sum_{i=1}^{N}\nabla \log p(x_i|\tilde{\theta}_j)\right\|^2$$

$$\leq \frac{3h^2\sigma^2}{4} + 3hd + \frac{3h^2}{4}\left\|\frac{N}{n}\sum_{i \in I_t}(\nabla \log p(x_i|\theta_j) - \nabla \log p(x_i|\tilde{\theta}_j)) + \sum_{i=1}^{N}\nabla \log p(x_i|\tilde{\theta}_j)\right.$$

$$\left. - \sum_{i=1}^{N}\nabla \log p(x_i|\theta_j) + \sum_{i=1}^{N}\nabla \log p(x_i|\theta_j)\right\|^2$$

$$\leq \frac{3h^2\sigma^2}{4} + 3hd + \frac{3h^2}{2}\left\|\sum_{i=1}^{N}\nabla \log p(x_i|\theta_j)\right\|^2$$

$$+ \frac{3h^2}{2}\left\|\frac{N}{n}\sum_{i \in I_t}(\nabla \log p(x_i|\theta_j) - \nabla \log p(x_i|\tilde{\theta}_j)) + \sum_{i=1}^{N}\nabla \log p(x_i|\tilde{\theta}_j) - \sum_{i=1}^{N}\nabla \log p(x_i|\theta_j)\right\|^2$$

$$\leq \frac{3h^2\sigma^2}{4} + 3hd + \frac{3N^2h^2\sigma^2}{2} + \frac{3N^2h^2\sigma^2}{n}.$$

The first inequality follows from Lemma 2 (with $r = 3$). The second inequality follows from the fact that $\|\nabla p(\theta)\|^2 \leq \sigma^2$ for all $\theta \in \mathbb{R}^d$ and the fact that $\eta_j \sim N(0, \sqrt{h})$. The third inequality again follows from Lemma 2 with $r = 2$. The last inequality follows from: (a) Lemma 2 with $r = N$ and (b) the bound in Equation 12. Substituting the bound in Equation 14, we get the following:

$$\mathbb{E}\left\|\theta_t - \tilde{\theta}_t\right\|^2 \leq m^2\left[\frac{3h^2\sigma^2}{4} + 3hd + \frac{3N^2h^2\sigma^2}{2} + \frac{3N^2h^2\sigma^2}{n}\right]. \tag{15}$$

Substituting Equation (15) in Equation 12 and substituting the minimum of the resultant bound and bound in Equation (13) into Equation 11 gives the desired result.

## B  Proof of Theorem 2

The proof of Theorem 2 is along the lines of Theorem 3. The key difficulty, in comparison to the analysis of SVRG-LD, comes from the fact that the full gradient is not computed after every few epochs. Again, we start with the following inequality proved by [3]:

$$\mathbb{E}(\phi - \bar{\phi})^2 \leq C\left(\frac{\frac{1}{T}\sum_t \mathbb{E}(\Delta V_t \psi(\theta_t))^2}{T} + \frac{1}{Th} + h^2\right). \tag{16}$$

for some constant $C > 0$. For SAGA-LD, we have the following inequality:

$$\frac{1}{C'T} \sum_t \mathbb{E}(\Delta V_t \psi(\theta_t))^2 \le \frac{1}{T} \sum_t \mathbb{E}[\|\nabla U_t(\theta_t) - \nabla U(\theta_t)\|^2]$$

$$= \frac{1}{T} \sum_t \mathbb{E} \left\| \frac{N}{n} \sum_{i \in I_t} (\nabla \log p(x_i|\theta_t) - \nabla \log p(x_i|\alpha_t^i)) + \sum_{i=1}^N \nabla \log p(x_i|\alpha_t^i) - \sum_{i=1}^N \nabla \log p(x_i|\theta_t) \right\|^2$$

$$= \frac{1}{Tn^2} \sum_t \mathbb{E} \left\| \sum_{i \in I_t} \left( N \left[ \nabla \log p(x_i|\theta_t) - \nabla \log p(x_i|\alpha_t^i) \right] - \left[ \sum_{i=1}^N \nabla \log p(x_i|\theta_t) - \sum_{i=1}^N \nabla \log p(x_i|\alpha_t^i) \right] \right) \right\|^2$$

$$= \frac{1}{Tn^2} \sum_t \mathbb{E} \sum_{i \in I_t} \left\| \left( N \left[ \nabla \log p(x_i|\theta_t) - \nabla \log p(x_i|\alpha_t^i) \right] - \left[ \sum_{i=1}^N \nabla \log p(x_i|\theta_t) - \sum_{i=1}^N \nabla \log p(x_i|\alpha_t^i) \right] \right) \right\|^2$$

$$\le \frac{1}{Tn^2} \sum_t \mathbb{E} \sum_{i \in I_t} \left\| N \left[ \nabla \log p(x_i|\theta_t) - \nabla \log p(x_i|\alpha_t^i) \right] \right\|^2 \le \frac{L^2 N}{Tn} \sum_t \sum_i \mathbb{E} \left\| \theta_t - \alpha_t^i \right\|^2 . \quad (17)$$

for some $C' > 0$. The first inequality is due to our assumption [A3] in Section 4. The third equality is obtained by using Lemma 1. The second inequality is due to the fact that $\mathbb{E}[\|\zeta - \mathbb{E}[\zeta]\|^2] \le \mathbb{E}[\|\zeta\|^2]$ for any random variable $\zeta \in \mathbb{R}^d$. The last inequality is follows from the Lipschitz continuity of $\nabla \log p(x_i|\theta)$ and uniform randomness of the set $I_t$.

Let $\gamma = 1 - (1 - 1/N)^n$. $\gamma$ represents the probability that an index is chosen at a particular iteration. Our goal is to bound the term $\sum_t \sum_i \mathbb{E} \left\| \theta_t - \alpha_t^i \right\|^2$. To this end, we observe the following:

$$\mathbb{E} \left\| \theta_t - \alpha_t^i \right\|^2 = \sum_{j=0}^{t-1} \mathbb{E} \left[ \mathbb{E} \left[ \left\| \theta_t - \alpha_t^i \right\|^2 \mid \alpha_t^i = \theta_j \right] \right]$$

$$\le \sum_{j=0}^{t-1} (t-j)^2 \left[ \frac{3h^2\sigma^2}{4} + 3hd + \frac{3N^2h^2\sigma^2}{2} + \frac{3N^2h^2\sigma^2}{n} \right] P(\alpha_t^i = \theta_j)$$

$$= \left[ \frac{3h^2\sigma^2}{4} + 3hd + \frac{3N^2h^2\sigma^2}{2} + \frac{3N^2h^2\sigma^2}{n} \right] \sum_{j=0}^{t-1} (t-j)^2 (1-\gamma)^{t-j-1} \gamma$$

$$= \left[ \frac{3h^2\sigma^2}{4} + 3hd + \frac{3N^2h^2\sigma^2}{2} + \frac{3N^2h^2\sigma^2}{n} \right] \gamma \sum_{j=1}^{t} j^2 (1-\gamma)^{j-1}$$

$$\le \left[ \frac{3h^2\sigma^2}{4} + 3hd + \frac{3N^2h^2\sigma^2}{2} + \frac{3N^2h^2\sigma^2}{n} \right] \gamma \sum_{j=1}^{\infty} j^2 (1-\gamma)^{j-1}$$

$$\le \frac{2}{\gamma^2} \left[ \frac{3h^2\sigma^2}{4} + 3hd + \frac{3N^2h^2\sigma^2}{2} + \frac{3N^2h^2\sigma^2}{n} \right]$$

$$\le \frac{8N^2}{n^2} \left[ \frac{3h^2\sigma^2}{4} + 3hd + \frac{3N^2h^2\sigma^2}{2} + \frac{3N^2h^2\sigma^2}{n} \right] . \quad (18)$$

The first equality is due to the law of total expectation. The first inequality can be obtained by using similar argument as the one in Equation (15). The second inequality follows from simple calculation of $P(\alpha_t^i = x_j)$. This in turn uses the fact that the set $I_t$ is selected uniformly randomly at each iteration. The last equality is due to the standard formula: $\sum_{j=1}^{\infty} j^2(1-\gamma)^{j-1} = (2-\gamma)/\gamma^3$. The last inequality is due to the following bound on $\gamma$:

$$\gamma = 1 - \left( 1 - \frac{1}{N} \right)^n \ge 1 - \frac{1}{1 + \frac{n}{N}} = \frac{n/N}{1 + n/N} \ge \frac{n}{2N}. \quad (19)$$

The first inequality in Equation (19) is due to the fact that $(1-x)^n \le 1/(1+nx)$ for $x \in [0,1]$ and $n \in N$, and the second last inequality is due to the fact that $n/N \le 1$. Substituting the bound in

Equation (18) into Equation (17), we have

$$\frac{1}{C'T} \sum_t \mathbb{E}(\Delta V_t \psi(\theta_t))^2 \leq \frac{L^2 N^3}{n^2} \left[ \frac{3h^2\sigma^2}{4} + 3hd + \frac{3N^2h^2\sigma^2}{2} + \frac{3N^2h^2\sigma^2}{n} \right]. \qquad (20)$$

Note that similar to Equation (13), the following bound holds for SAGA-LD:

$$\frac{1}{C'T} \sum_t \mathbb{E}(\Delta V_t \psi(\theta_t))^2 \leq \frac{1}{Tn^2} \sum_t \mathbb{E} \sum_{i \in I_t} \left\| N \left[ \nabla \log p(x_i|\theta_t) - \nabla \log p(x_i|\alpha_t^i) \right] \right\|^2 \leq \frac{2N^2\sigma^2}{n}$$
$$(21)$$

Using the minimum of the bounds in Equation (21) and (20) in Equation (16) gives us the desired result.

## C  Other Lemmatta

We state few useful and well-known lemmas in this section.

**Lemma 1.** *If random variables $z_1, \ldots, z_r$ are independent and have mean 0, then*

$$\mathbb{E}\left[\|z_1 + \ldots + z_r\|^2\right] = \mathbb{E}\left[\|z_1\|^2 + \ldots + \|z_r\|^2\right].$$

*Proof.* We have the following:

$$\mathbb{E}\left[\|z_1 + \ldots + z_r\|^2\right] = \sum_{i,j=1}^{r} \mathbb{E}\left[z_i z_j\right] = \mathbb{E}\left[\|z_1\|^2 + \ldots + \|z_r\|^2\right].$$

The second equality follows from the fact that $z_i$'s are independent and have mean 0. $\qquad \square$

**Lemma 2.** *For random variables $z_1, \ldots, z_r$, we have*

$$\mathbb{E}\left[\|z_1 + \ldots + z_r\|^2\right] \leq r\mathbb{E}\left[\|z_1\|^2 + \ldots + \|z_r\|^2\right].$$

## D  SVRG-LD

The memory complexity for SAGA-LD is high because the approximate gradient $g_\alpha$ is updated at each step. This can be avoided by fully updating the gradient every $m$ iterations in one expensive evaluation, and using that gradient $\tilde{g}$ as an approximation to the true gradient for the next $m$ steps. Concretely, every $m$ passes through the data, we evaluate the gradient on the entire data set, to obtain an estimate $\tilde{g} = \sum_{i=1}^{N} \tilde{g}_i$, where $\tilde{g}_i = \nabla \log p(x_i|\tilde{\theta})$ is the local gradient evaluated at the current time.

As we iterate through the data, if a data point is not selected in the current minibatch, we approximate its gradient with $\tilde{g}_i$. If $I_t = \{i_{1t}, \ldots i_{nt}\}$ is the minibatch selected at iteration $t$, then we approximate $\sum_{i=1}^{N} \nabla \log p(x_i|\theta_t)$ so that

$$\sum_{i=1}^{N} \nabla \log p(x_i|\theta_t) \approx \frac{N}{n} \sum_{i \in I_t} \left(\nabla \log p(x_i|\theta_t) - \tilde{g}_i\right) + \tilde{g}, \qquad (22)$$

This yields an update of the form

$$\delta\theta_t = \frac{h_t}{2} \left( \nabla \log p(\theta_t) + \frac{N}{n} \sum_{i \in I_t} \left(\nabla \log p(x_i|\theta_t) - \tilde{g}_i\right) + \tilde{g} \right) + \eta_t \qquad (23)$$

where $\eta_t \sim N(0, h_t)$. Pseudocode for this procedure is given in Algorithm 2 (and repeated in this supplement as Algorithm 3).

As with SAGA-LD, we note that the update in Equation 7 (and Equation 23) gives an unbiased estimate of the true gradient, since $I_t$ is chosen uniformly at random (with replacement) from $[N] =$

**Algorithm 3:** SVRG-LD

1: **Input:** $\tilde{\theta} = \theta_0 \in \mathbb{R}^d$, epoch length $m$, step sizes $\{h_t > 0\}_{i=0}^{T-1}$
2: **for** $t = 0$ **to** $T - 1$ **do**
3:     **if** ($t$ mod $m = 0$) **then**
4:         $\tilde{\theta} = \theta_t$
5:         $\tilde{g} = \sum_{i=1}^{N} \nabla \log p(x_i | \tilde{\theta})$
6:     **end if**
7:     Uniformly randomly (with replacement) pick a set $I_t$ from $\{1, \ldots, N\}$ such that $|I_t| = n$
8:     Randomly draw $\eta_t \sim N(0, h_t)$
9:     $\theta_{t+1} = \theta_t + \frac{h_t}{2} \left( \nabla \log p(\theta_t) + \frac{N}{n} \sum_{i \in I_t} (\nabla \log p(x_i | \theta_t) - \nabla \log p(x_i | \tilde{\theta})) + \tilde{g} \right) + \eta_t$
10: **end for**
11: **Output:** Iterates $\{\theta_t\}_{t=0}^{T-1}$

---

$\{1, \ldots, N\}$, since we have $\mathbb{E}[\frac{N}{n} \sum_{i \in I_t} \tilde{g}_i - \tilde{g}] = 0$. Therefore, the term $\frac{N}{n} \sum_{i \in I_t} \tilde{g}_i - \tilde{g}$ does not add any bias to the stochastic gradient.

By calculating the full gradient after every $m$ iterations, we ensure that the accuracy of the approximation is not allowed to decrease too significantly, and ensure that the variance of the updates is controlled. We provide concrete bounds in Section 4. We note that, if $m \geq \lfloor N/n \rfloor$, the computational complexity of SVRG-LD is similar to SGLD.

One desirable property of SVRG-LD is that it has low memory requirements: SVRG requires just $O(d)$ extra memory (in order to store the approximate gradient $\tilde{g}$), in comparison with SGLD. However, there is a potentially large computational burden due to the need to periodically calculate the full gradient. In practice, we found that computation was a greater bottleneck in the examples considered.

# E    Other Experiment Results

Figure 4: Performance comparison of SGLD and SAGA-LD on the regression task. The x-axis and y-axis represent the number of pass through the entire data and average test MSE respectively. Please refer to the section 5.1 for details