[Reviews · NeurIPS 2016]

Reviewer 1

Summary

Stochastic Gradient MCMC algorithms such as Stochastic Gradient Langevin Dynamics (SGLD) are becoming increasingly popular as a way to perform Bayesian inference efficiently in large datasets. The SGLD algorithm is very similar to the SGD optimization algorithm, with the update equation being just the SGD update + injected Gaussian noise. The variance of the gradient at each step (from using a mini-batch instead of the whole dataset) adversely affects the convergence of the SGLD algorithm. Since the same problem affects SGD, there has been recent work in the optimization community on techniques for reducing this variance. In this paper, the authors apply one of these techniques, the SAGA algorithm, to improve the convergence of SGLD. The algorithm keeps the gradient of each datapoint, and their sum in memory. This sum of gradients of all datapoints is used to make updates in every iteration. However, after the first iteration, only gradients in a minibatch are computed, so the sum of gradients is a little stale as it involves gradients computed at parameter values from previous iterations. But the computational complexity is the same as SGLD, and using the sum of gradients over the whole dataset results in lower variance, which leads to faster convergence and lower asymptotic error. The basic algorithm is memory heavy as it requires storing the gradient of each datapoint, so the authors also propose a variant of the algorithm to address this, at the cost of more computation.

Qualitative Assessment

I have one key concern, which may be a misunderstanding on my part as I did not check the supplementary section in detail. The update at each time step is computed using gradients of different parameter values (theta), some of which were generated arbitrarily many time steps ago. This dependence on previous samples means that the SAGA-LD chain is not Markov. The proofs seem to be based on a result for SG-MCMC chains, but I am not sure if the result easily applies to SAGA-LD because of the violation of the Markov property. Other than the above point, I think this is a very useful line of work. Reducing the variance of SG-MCMC is very important to accelerate convergence. This also leads to better mixing which results in estimates with lower variance. Moreover, the same technique seems applicable to many other SG-MCMC algorithms. Clarity: The paper is relatively well written and clear. Novelty: The paper applies variance reduction techniques from the optimization literature to similar SG-MCMC. So, although not very original, no one has done this, and it is an important angle to investigate. Other Comments/Questions: - I think in Figs. 1 and 4, SGLD should be run until convergence. Although SAGA-LD converges faster, does it get to the same error or is there a higher bias? - In Fig 3, what do the variance plots mean? i.e. variance of what quantity is being measured? - Do you have an intuition as to why there are spikes in the some of the test MSE curves for SAGA-LD after every pass of the data. e.g. the twitter and music plots in Figure 4 of the supplementary. - Algorithm 1, Line 4, does this have to be sampling *with replacement*? Sampling without replacement has lower variance, but I am not sure if this will affect your proofs. - The sentences around Eqn 3 such as Poisson equation, generators etc may not be well known in the ML community, so it would be nice to add more explanation or references. - Also mention the expansion of SAGA somewhere in the paper. The experiments do an adequate job of showing that SAGA-LD improves on SGLD. But I think the paper will be strengthened if it also shows comparisons to optimization algorithms like SGD, and use applications where being Bayesian is advantageous. Many ML practitioners today are, in general, content with SGD and other optimization algorithms. Convincing applications showing that there are advantages to being Bayesian will make your paper appeal to a wider audience beyond the SG-MCMC community. This can be e.g. lower error from posterior averaging, better uncertainty estimates etc.

Confidence in this Review

2-Confident (read it all; understood it all reasonably well)


Reviewer 2

Summary

This paper proposes two stochastic gradient sampling methods, motivated by combining the approach of Stochastic Gradient Langevin Dynamics with recent variance reduction techniques from the stochastic approximation literature.

Qualitative Assessment

The overall structure of the paper is reasonably clear, and the algorithms themselves are stated reasonably precisely (although there are inconsistencies in the notation used within the algorithm boxes), with clear motivation from the introductory material. There are some typos (see minor points below), and quite a few instances where notation is used without being defined – in some circumstances it is clear what is meant, but in several circumstances it impeded my understanding of the paper. For example, the quantities \bar{\phi} and \hat{\phi} that appear in Theorems 1, 2, and 3 are not defined. An important point which I don’t believe has been picked up in the paper is that the newly proposed methods are not MCMC methods as claimed in the abstract - the fact that gradients are stored between timesteps breaks the Markovian structure of the process. In relation to the “Analysis” section of the paper, the two main theoretical results supporting the algorithms are clearly stated (although as noted above some notation is undefined). There are some particularly strong conditions in assumption [A2], particularly the condition on uniform boundedness of the gradient log-prior and log-likelihoods, which restricts the applicability of Theorems 2 and 3. The condition rules out (for example) Gaussian priors or likelihoods, and therefore the theorems subsequently proven do not apply to any of the models in the experiments. It would be useful to explain what kinds of models the theoretical results can be applied to, and what kinds of models are excluded by these strong conditions. I found the proofs of Theorems 2 and 3 in the appendix difficult to follow, due to undefined notation (e.g. C^\prime appearing in Eqn (16)), and missing notation (presumably the norms appearing in the displayed equations between lines 329 and 330 should have expectations taken over them). Additionally, there appears to be a mistake in the displayed equations between lines 329 and 330; the expected value of the squared norm of eta_j, a d-dimensional multivariate normal random variable (with mean vector 0 and covariance matrix \sqrt{h} I), appears to be bounded by h. In fact, the expected value is exactly hd, where d is the dimension of the variable. A general point about the final paragraph of Section 4 is that the bounds obtained in Theorems 1, 2, and 3 all have 1/(Th) and h^2 terms in common - would this mean that the bounds all behave similarly asymptotically, even if the first term in the bounds of Theorems 2 and 3 decays quicker than that of Theorem 1? If this is the case, then the claim that convergence of the MSE of SAGA-LD and SVRG-LD is significantly faster than that of SGLD might not hold. In relation to the “Experiments” section of the paper, there is a good range of experiments (Bayesian linear regression, logistic regression, and ICA) which clearly demonstrate that SAGA-LD performs well in comparison to SGLD in several circumstances. It is a shame that no experiments relating to the SVRG-LD method were included. The SAGA-LD algorithm begins with a full pass through the data to calculate the gradient log posterior at the initial parameter value - would this materially affect the comparisons drawn for the experiments if the methods were compared on a wall-clock basis, rather than number of passes through the data? As a minor note, it appears as though in several of the regression and classification experiments, SAGA-LD and SGLD reach a similar level of MSE after several passes through the data set. It would be interesting to see whether initialising both algorithms at the MAP estimate of the posterior (say) would reduce the difference in performance between the two methods. Comparing the performance of the sampling schemes with pure optimisation algorithms such as SGD/SAGA could also provide a useful benchmark for the results obtained. The mixture model experiment only shows results for SGLD (although perhaps this is a typo in line 258?), so it is not clear to me why it has been included in the paper. Minor points: Throughout: “Robbins-Monroe” should be “Robbins-Monro” Line 23: the word “method” is repeated Line 50: “success” should be “successes” Line 76: U is the unnormalized *negative* log posterior Eqn (3): In the definition of the generator, the expectation should be conditioned on \theta_0 = \theta References: The 1951 Robbins-Monro paper appears twice in the list of references Appendix Section E: the “protein” graph appears twice

Confidence in this Review

3-Expert (read the paper in detail, know the area, quite certain of my opinion)


Reviewer 3

Summary

The authors propose the use of two known reduced-variance estimators of the stochastic gradient (SAGA and SVRG), in order to accelerate the convergence of SGLD. Several examples are presented.

Qualitative Assessment

The paper has an interesting idea and examples. A downside of the paper is that it does not present conparisons between the mixing speeds at equilibrium (via ACF or ESS). typos: L59 conversion -> convergence 279 generic enough (very cannot go with enough)

Confidence in this Review

2-Confident (read it all; understood it all reasonably well)


Reviewer 4

Summary

The paper applies the method of variance reduction in SGD to SGLD, and shows improved convergence in theory as well as improved performance in practice.

Qualitative Assessment

The paper is written clearly and easy to follow. It is good to see a useful idea of SGD can be borrowed to SGLD with provable convergence improvement. The result is not surprising, but it is good that the author explored it and found it useful both theoretically and practically. Both Theorem 2 and Theorem 3 are based on fixed step-size analysis. Theorem 2 of the paper shows that SAGA-LD improves the MSE when h < n^2/N^2 \sigma^2, or when h < n/N/L. However in the experiment, a decayed step size is used. Also the advantage seems to show in the stage when h is still fairly large. So it is a bit unclear if the improvement is due to the MSE of the Bayesian estimate is reduced with the variance reduction scheme. It would be good to include the analysis of decayed step-size algorithm as was done in [3]. Minor comments: 1. The title contains a typo: "Langevian" -> "Langevin".

Confidence in this Review

3-Expert (read the paper in detail, know the area, quite certain of my opinion)


Reviewer 5

Summary

This article addresses methods of variance reduction in Stochastic Gradient Langevin Dynamics. It is well known that SGLD has significant issues, e.g. the reliance on a diminishing sequence of stepsizes (necessary for convergence theory), and, in the case where a fixed stepsize is used, slow convergence to the invariant distribution. Many efforts are being made to address these failings and to enhance the performance of the method. The current article aims to use difference approximations to effectively reduce the variance in SGLD simulations.

Qualitative Assessment

I think the paper is interesting and should definitely appear in NIPS. The general idea being explored here is not entirely new. For example the authors should definitely cite Shang et al, Covariance-Controlled Adaptive Langevin Thermostat for Large-Scale Bayesian Sampling, NIPS, 2015, which uses similar covariance approximations to accelerate convergence (indeed it would be a good comparison for the method of this article). I like the mathematical treatment and the numerical experiments are well presented.

Confidence in this Review

2-Confident (read it all; understood it all reasonably well)